# Machine Learning Algorithm: Texture Analysis in CNO and Application in Distinguishing CNO and Bone Marrow Growth-Related Changes on Whole-Body MRI

**DOI:** 10.3390/diagnostics14010061

**Published:** 2023-12-27

**Authors:** Marta Forestieri, Antonio Napolitano, Paolo Tomà, Stefano Bascetta, Marco Cirillo, Emanuela Tagliente, Donatella Fracassi, Paola D’Angelo, Ines Casazza

**Affiliations:** 1Imaging Department, Bambino Gesù Children’s Hospital, IRCCS, 00165 Rome, Italy; paolo.toma@opbg.net (P.T.); stefano1.bascetta@opbg.net (S.B.); paola.dangelo@opbg.net (P.D.); ines.casazza@opbg.net (I.C.); 2Medical Physics Department, Bambino Gesù Children’s Hospital, IRCCS, 00165 Rome, Italy; antonio.napolitano@opbg.net (A.N.); emanuela.tagliente@opbg.net (E.T.); donatella.fracassi@opbg.net (D.F.)

**Keywords:** whole-body magnetic resonance imaging, chronic non-bacterial osteomyelitis, bone marrow, machine learning, texture analysis, children

## Abstract

Objective: The purpose of this study is to analyze the texture characteristics of chronic non-bacterial osteomyelitis (CNO) bone lesions, identified as areas of altered signal intensity on short tau inversion recovery (STIR) sequences, and to distinguish them from bone marrow growth-related changes through Machine Learning (ML) and Deep Learning (DL) analysis. Materials and methods: We included a group of 66 patients with confirmed diagnosis of CNO and a group of 28 patients with suspected extra-skeletal systemic disease. All examinations were performed on a 1.5 T MRI scanner. Using the opensource 3D Slicer software version 4.10.2, the ROIs on CNO lesions and on the red bone marrow were sampled. Texture analysis (TA) was carried out using Pyradiomics. We applied an optimization search grid algorithm on nine classic ML classifiers and a Deep Learning (DL) Neural Network (NN). The model’s performance was evaluated using Accuracy (ACC), AUC-ROC curves, F1-score, Positive Predictive Value (PPV), Mean Absolute Error (MAE) and Root-Mean-Square Error (RMSE). Furthermore, we used Shapley additive explanations to gain insight into the behavior of the prediction model. Results: Most predictive characteristics were selected by Boruta algorithm for each combination of ROI sequences for the characterization and classification of the two types of signal hyperintensity. The overall best classification result was obtained by the NN with ACC = 0.91, AUC = 0.93 with 95% CI 0.91–0.94, F1-score = 0.94 and PPV = 93.8%. Between classic ML methods, ensemble learners showed high model performance; specifically, the best-performing classifier was the Stack (ST) with ACC = 0.85, AUC = 0.81 with 95% CI 0.8–0.84, F1-score = 0.9, PPV = 90%. Conclusions: Our results show the potential of ML methods in discerning edema-like lesions, in particular by distinguishing CNO lesions from hematopoietic bone marrow changes in a pediatric population. The Neural Network showed the overall best results, while a Stacking classifier, based on Gradient Boosting and Random Forest as principal estimators and Logistic Regressor as final estimator, achieved the best results between the other ML methods.

## 1. Introduction

Chronic non-bacterial osteomyelitis (CNO) is a chronic inflammatory bone syndrome disease caused by a dysregulation of the immune system [1,2,3] that most frequently affects children and adolescents, although it is not exclusively a pediatric disease [3,4].

The clinical onset occurs in the age group between 7 and 12 years old [5]; girls are more frequently affected (F:M = 2:1) [6,7]. Diagnosis is uncommon before the age of 3 years old; all ethnic groups can be involved [8]. Due to the non-specificity of the symptoms and the subtle clinical presentation, diagnosis is often difficult and delayed by about two years [9]. The clinical spectrum of chronic non-bacterial osteomyelitis can range from a single and self-limiting bone localization to chronic and recurrent multifocal localizations (CRMO). However, because the disease may onset or remain unifocal, the use of the term CNO has to be preferred [8]. Sites most frequently involved are the metaphysis of the long bones, the pelvis, the spine, the shoulders, the clavicles and the jaw [10,11]. Currently, CNO remains a diagnosis of exclusion due to the lack of specific features and absence of validated diagnostic criteria. However, there are clinical scores such as the Jansson and Bristol criteria that can be supportive in correct etiologic framing when there is diagnostic suspicion of CNO [12,13]. Biopsy of bone lesions is generally performed to exclude other diseases with similar presentation [14]. Anatomo-pathological evaluation shows non-specific inflammation in the absence of infection [12]. Conventional radiography has a low sensitivity (about 80% false negatives) in detecting inflammatory bone lesions in the early stage of the disease [15]. Hyperostosis and periosteal reaction may be present, without the detachment of the periosteum or the formation of subperiosteal bone sequesters, characteristic elements of bacterial osteomyelitis [5,11].

Although bone scintigraphy (performed using methylene diphosphonate or hydroxymethylene diphosphonate chelated with technetium-99m) is a good tool for identifying symptomatic and non-symptomatic sites of inflammation, studies have shown that its sensitivity in detecting the precise number of bone lesions is lower than that of magnetic resonance [16].

The gold standard for initial evaluation and disease follow-up in CNO is the Whole-Body MRI (WBMRI), which allows detection of the presence of hyperintense edema-like lesions on the short tau inversion recovery (STIR) sequences. Their morphology is variable; therefore, it is possible to detect signal alterations ranging from poorly defined ones to confluent ones [17]. Some authors refer to them as “field in flames” [18] (Figure 1a). However, the high signal on STIR sequences is not specific because it may simply represent increased water content [19,20] compared with the surrounding bone; for this reason, it is not specific for CNO, and it can also be detected in other pathologic conditions (such as infectious osteomyelitis, leukemia, Langerhans cell histiocytosis, scurvy, etc.) or in a physiological contest, such as in the presence of hematopoietic bone marrow, well represented in the bones of pediatric patients [21]. Red marrow is made up of 40% water, 40% fat and 20% protein. Therefore, it appears hyperintense on T2-weighted (T2w) and fluid-sensitive sequences [22,23] (Figure 1b). Extracting clinically relevant information from radiological imaging is proving to be a powerful tool in the implementation of artificial intelligence algorithms [24]. Indeed, radiomics is commonly used to predict patient outcome through automated high-throughput feature extraction, and, particularly, the main purpose is to elucidate the subtle relationships between the characteristics of the image and the disease status. Recent studies have shown a performance level almost comparable to that of practicing radiologists [25,26,27]. In the literature, there are examples of automatic segmentation of the hyperintense signal of CNO and bone marrow [28], but it still lacks studies that focus on the differentiation between the two types of signal through ML. In addition, we adopt the unified framework Shapley additive explanations (SHAP), a technique commonly used to interpret models, rank the feature importance, identify which features contribute negatively or positively to the prediction performance and also further select the optimal feature sets [29].

Starting from these assumptions, the purpose of our study is to analyze the texture characteristics of CNO bone lesions, identified as areas of altered signal intensity on STIR sequences, and to distinguish them from the hyperintense signal of red bone marrow through ML. We have attempted to provide clinical help for the diagnosis of CNO in order to avoid overdiagnosis [30] and consequent overtreatment of the pathology, and to limit invasive procedures such as bone biopsy.

## 2. Materials and Methods

The present retrospective study included pediatric patients who underwent WBMRI exams at the Bambino Gesù Pediatric Hospital (OPBG) in Rome. We obtained information about age, gender and clinical-laboratoristic and anatomo-pathological characteristics through the clinical computerized system (clinical OBG, Bambino Gesù/Roma, Italy); the radiological assessments were obtained through the radiological information system (RIS: Fenix EL.CO/Milano, Italy), and the images were selected by the picture archiving and communication system (PACS Cararestream/Rochester, NY, USA).

### 2.1. Patients 

We included 66 pediatric patients with diagnosis of CNO who had undergone at least one WBMRI examination showing the presence of unifocal or multifocal inflammatory localizations. We considered all the new-onset lesions that appeared during the follow-up period for each patient. To be able to make a comparison between CNO lesions and healthy bone (control group) that were superimposable in terms of location, extension and morphology through ROI, symmetrical lesions were excluded; only unilateral ones were considered in order to use the contralateral bone, free from signal alterations, as a term of comparison. Then, we selected 28 patients diagnosed with dermatomyositis (without skeletal or articular involvement) who underwent Whole-Body MRI examination. This latter group of patients was used to sample and to extrapolate texture data of the bone marrow growth-related changes. In the absence of histological evidence, for ethical reasons, areas of hyperintensity in the context of a healthy bone were interpreted as expression of physiological changes in bone marrow appearance related to growth in the presence of certain characteristics such as congruence with age, topographical distribution, pattern and symmetry, taking into account the notions known from older studies to the most recent bone marrow identification methods [22,31,32].

Summarizing, we created two distinct databases: the first one containing patients with CNO (which we labeled “CNO”) and the second one containing the group of patients from which we extracted the characteristics of the red bone marrow (that we labeled “red marrow”).

Patients with exclusively symmetrical inflammatory skeletal lesions and patients with concomitant neoplastic and infectious pathologies were excluded from the study. Patients who had undergone WBMRI with DIXON T2w sequences in place of STIR were excluded too. Only patients who had undergone examinations in the same MRI scanner were included. Informed consent for all patients was obtained from their legal guardian. The study was approved by the Ethics Committee of the Bambino Gesù Pediatric Hospital (protocol no. 2268_OPBG_2020) and performed in accordance with the Declaration of Helsinki.

### 2.2. Imaging 

All WBMRIs were performed on a Siemens Aera 1.5 T Device, NUMARIS system software, version 4 (syngo MR E11). For each patient, we applied the Whole-Body protocol, with the acquisition of STIR coronal images using the following parameters: TE, 58 ms; TR, 5000 ms (±500 ms); TI, 160 ms; flip angle, 150°; voxel matrix size, 1.2 × 1.2 × 3 mm. All the examinations included the entire body volume, from head to toe, using an additional sagittal plane on the spine.

The mean duration of each WBMRI examination was 40 ± 5 min.

### 2.3. Processing of Images and Extraction of Features

Regions of interest (ROIs) were selected manually for each patient using the opensource 3D Slicer program version 4.10.2 (https://www.slicer.org/ accessed on 23 October 2023). Three groups of ROIs were selected from the “CNO” database (lesion, contralateral bony segment, healthy bone on the affected bony segment); two groups of ROIs were selected from the “red marrow” database (marrow, healthy). All the masks from which the ROIs were extracted and the ROIs themselves have been saved in .nii format.

Concerning the “CNO” patients, the first volumetric ROIs were obtained by selecting the regions in which the inflammatory bone lesion was viewable for each slice in accordance with the radiologist’s report and the biopsy results. We called these regions of interest “lesion” (“lesion1”, “lesion2”, etc.) (Figure 2). The second ROIs were sampled in the healthy contralateral bone for the even bones, and on the bone portions without hyperintense signal for odd skeletal segments, sampling the same volume share of the “lesions” ROIs. We classified these ROIs as “contralateral” (“contralateral1”, “contralateral2”, etc.). The third ROIs were plotted on a portion of bone considered radiologically healthy and were used for the calibration of the previous measurements (“healthy1”, “healthy2”, etc.).

Concerning the “red marrow” group, the first ROIs were sampled on the hyperintense regions of the bone represented by bone marrow, such as the metaphyseal regions of the appendicular bones, and were classified as “marrow” (“marrow1”, “marrow2”, etc.) (Figure 3). The second ROIs were sampled on the bone regions without hyperintense signal, in particular, in the regions where the bone marrow conversion process was already completed, for the calibration of the previous ROIs and were classified as “healthy” (“healthy1”, “healthy2”, etc.).

We carried out the “intensity non-standardness” correction to standardize our data through MATLAB R2017a software.

Afterwards, we extracted the radiomic features of shape, intensity and texture (GLCM, GLDM, GLSZM, GLRLM, NGTDM) using the Pyradiomics package on Python 2.7. The characteristics were extracted from original images and filtered (wavelet decomposition, Laplacian of Gaussian, exponential, logarithmic and gradient).

We also implemented the fractal dimension features: 2D box counting, 3D box counting and differential.

### 2.4. Features Selection and Classifications

The Image Biomarker Standardization Initiative (IBSI) manual has been a useful reference for the development of robust radiomics extraction and analysis [33]. In fact, we carried out the intensity non-standardness correction to standardize our data through MATLAB R2017a software. In addition, we normalized the entire dataset to a fixed intensity range ([0,255]), as suggested by the guide. Afterwards, we extracted the radiomic features of shape, intensity and texture (GLCM, GLDM, GLSZM, GLRLM, NGTDM) using the Pyradiomics package on Python 2.7. The characteristics were extracted from original images and filtered (wavelet decomposition, Laplacian of Gaussian, exponential, logarithmic and gradient). The extracted feature distribution was standardized by taking out outliers, removing the mean and scaling it to unit variance with the Python Standard Scaler package. A feature selection tool was applied in order to remove features with less importance and reduce the complexity of the prediction. For this purpose, we used the Boruta algorithm, a powerful feature selector method, which trained a Random Forest Classifier on a duplicated dataset (composed by original and shadow features) and marked a feature as important, comparing its Z-scores with that of the duplicate [34]. Definitely, starting from 783 features extracted by Pyradiomics, the Boruta algorithm selected 99 final most predictive features. The Python module which implemented this function was the boruta_py module, freely accessible from the GitHub repository (https://github.com/scikit-learn-contrib/boruta_py accessed on 23 October 2023). With the aim of maximizing the performance and robustness of the ML model, an optimization search grid algorithm was applied on nine classical ML classifiers (Figure 4), including ensemble and non-ensemble learners—AdaBoost (AB), Extreme Gradient Boosting (xGB), Gradient Boosting (GB), Decision Tree (DT) and Random Forest (RF), Logistic Regressor (LR), two Stacking classifiers (ST, ST_ABC) and KNeighbors (KN)—and a Neural Network (NN). These algorithms were chosen based on their known performances and extensive use in the literature [26]. AB and GB are boosting methods, where AB reweights instances and GB excels with categorical data. xGB is highlighted as the fastest implementation of gradient-boosted trees. RF is presented as a collection of decision trees for object classification based on attributes. DT serves as a basic algorithm for classification. LR is a widely used linear classifier for capturing linear relationships in data. Stacked Generalization in ST and ST_ABC is employed as an ensemble algorithm combining predictions from GB, RF and LR. k-Nearest Neighbors (kNN) relies on data space distance for classification. In particular, Extreme Gradient Boosting (xGB) is specifically implemented using the XGBoost package, while other classifiers are part of the Scikit-learn package. The NN, implemented using the Keras library, is specially designed for binary classification tasks with 2 hidden layers and an output layer with a sigmoid activation; L2 regularization and dropout are applied in the hidden layers to prevent overfitting (Figure 5). To compare the chosen classifiers, the accuracy (ACC), ROC-AUC curves, F1-score, Positive Predictive Value (PPV), Mean Absolute Error (MAE) and Root-Mean-Square Error (RMSE) were evaluated on 50 different splits of training and test data due to 10 times K-fold cross-validation (K = 10) computing. The data from K-fold cross-validation were used to statistically test the differences in performance due to each of the used classifiers. An independent sample T-test was exploited to statistically test this difference. A Bonferroni correction was also applied to make the correction for multiple comparison. Due to the unbalanced nature of the dataset, the Synthetic Minority Oversampling Technique (SMOTE) was implemented, introducing new synthesized samples from the existing ones in the minority class [35]. Once optimal parameters were assessed, model performances were evaluated in term of F1-score, Precision and AUC-ROC curve with 95% CI [36]. The literature reports AUC-ROC curves, which show the trade-off between false-positive and true-positive rates in the classification, as the better choice of measure in comparing learning algorithms [35]. The evaluation of Mean Absolute Error (MAE) and Root-Mean-Square Error (RMSE) was used to statistically compare the performance of different classifiers. Furthermore, we used SHAP to gain insight into the behavior of the prediction model. Due to the high dimensionality of radiomic features extracted, to improve interpretability of the model trained, we computed the SHAP value, which is a novel framework based on the game theory [29]. Specifically, SHAP implementation quantifies the contribution of each feature to predict the outcome using the XGBoost model with the highest predictive parameters selected by the grid search algorithm. This framework allowed us to verify whether each feature contributed positively or negatively to predict the outcome. Figure 6 shows the SHAP reporting of the 20 ranking significative features according to the global and local point of view of the method adopted.

### 2.5. Results

We included 66 pediatric patients (23 males and 43 females) with a confirmed diagnosis of non-bacterial osteomyelitis who underwent a bone biopsy to exclude other pathologies and a WBMRI examination at the Bambino Gesù Pediatric Hospital in a period between January 2016 and June 2021. The mean age at diagnosis was 10.9 ± 3.8 years old. A total of 240 ROIs were sampled on CNO lesions.

We also included 28 patients (12 males and 16 females), mean age 11.2 ± 5 years old, with a diagnosis of dermatomyositis and free from bone pathology undergoing WBMRI examination; 87 ROIs were identified on areas of hyperintensity and interpreted as “red bone marrow” based on morpho-structural characteristics, congruence with age, topographical distribution and symmetry.

After selecting the regions of interest from both groups, we performed the texture analysis. Shape, intensity and 75 texture characteristics (GLCM, GLDM, GLSZM, GLRLM, NGTDM) were extracted from original and filtered images (wavelet decomposition, Gaussian Laplacian, exponential, logarithmic and gradient).

The ML analysis led to the results shown in Table 1 according to Accuracy, ROC-AUC with 95% confident interval (CI), F1-score, PPV, MSE and RMSE assessment.

The statistical tests among the ML classifiers’ ACC performances were not significant, with the following exceptions (*p* < 0.05): KN vs. GBx; KN vs. BG; RF vs. KN; LR vs. KN; ST vs. KN; KN vs. AB; ST_ABC vs. KN.

The statistical tests among the classifiers’ F1-score performances were not significant, with the following exceptions (*p* < 0.05): KN vs. GBx; KN vs. BG; RF vs. KN; ST vs. KN; KN vs. AB; ST_ABC vs. KN. 

The statistical tests among the classifiers’ Positive Predictive Value performances were not significant, with the following exceptions (*p* < 0.05): KN vs. GBx; KN vs. BG; RF vs. KN; ST vs. KN. 

The statistical tests among the classifiers’ ROC-AUC performances were not significant, with the following exceptions (*p* < 0.05): KN vs. GBx; KN vs. BG; RF vs. KN; LR vs. KN; ST vs. KN. 

Box plots comparing classification performances are reported in Figure 7.

## 3. Discussion

A hyperintense signal on sensitive fluid sequences is the main MRI characteristic detectable in patients with CNO. However, the same type of signal can be found in numerous other conditions such as transient osteoporosis, avascular necrosis, metastatic or primary bone lesions, infectious osteomyelitis, reflex sympathetic dystrophy, trauma, micro traumatism and biomechanical alterations [37]. The distinction between a pathological and a physiological hyperintensity can represent a challenge for the radiologist.

The main purpose of our study was to use ten automatic learning models to discriminate two types of hyperintense signal on STIR sequences: the inflammatory CNO one and the bone marrow one. Red bone marrow produces a signal hyperintensity that can vary from mildly to moderately increased to a fluid-like signal [31] that can mimic many pathological conditions [21]. The purpose of our study was to implement an automatic recognition system that also assists the eye of radiologists.

Pal et al. have revealed the common presence of multiple areas of high signal on STIR sequences of the feet of pediatric patients, sites frequently involved in CNO, concluding that those aspects represent a normal evolution of the growing bone and not necessarily a pathological sign [38]. In a more recent study, Shabshin et al. come to the same conclusions by assigning a meaning of normality to the hyperintensities frequently found in the tarsus and pelvic bones of children which corresponded to hematopoietic marrow residue [37].

The recent literature shows a growing interest in the use of Machine Learning methods in the study of musculoskeletal pathologies. In particular, there are studies concerning the ability to distinguish between benign or malignant vertebral fractures [39], skeletal maturity studies [40] and studies concerning the distinction between cartilaginous bone tumors [41].

There are few studies currently available in the literature concerning the analysis through radiomics of the hyperintensities of the “edema-like” signal on fluid-sensitive sequences. Among them, the study of Faleiros et al., in which Machine Learning analysis is used to classify active sacroiliitis on magnetic resonance STIR images, showed an agreement between the radiologist’s diagnosis and the algorithm of greater than 90% [42]. Furthermore, Kepp et al. show the superiority of the quantitative approach over the qualitative one in the discrimination of inflammatory from degenerative changes in the sacroiliac joints [43].

In addition to classic images, the new diagnostic technologies produce an enormous wealth of numerical data that simple visual observation (the so-called qualitative analysis) is unable to process. The “Radiomics” process consists of the extraction and analysis of advanced quantitative imaging features with the aim of creating extractable databases to conduct predictive and prognostic correlations between images and patient outcome [44,45].

In the context of Machine Learning approaches, the Support Vector Machine (SVM) is a method that uses hyperplanes to separate the supplied samples in an optimal way. This method transforms a multi-solution problem into a single-solution problem [46,47]. Submitting the classifiers to repeated choices, it is possible to train them in the recognition of certain variables [48].

To the best of our knowledge, the stream of literature that focuses on inflammatory CNO lesions lacks studies that implement quantitative methods for distinguishing the pathological from the physiological signal hyperintensities, and the results we obtained are encouraging because six classifiers out of nine achieved an accuracy of greater than 80% in the discrimination of the two types of hyperintensity.

The Deep Learning Neural Network achieved the overall best results (ACC = 0.91, AUC = 0.93 with 95% CI 0.91–0.94, F1-score = 0.94, PPV = 93.8%), confirming NNs’ robust performances and adaptability in image classification, with the drawback of being extremely sensitive to parameter overfitting especially with small-scale data samples.

Among the classical ML classifiers, the Stacking classifier (ST) achieved the best results (ACC = 0.85, AUC = 0.81 with 95% CI 0.8–0.84, F1-score = 0.9, PPV = 90%). ST is an ensemble learner in which, compared to the bagging ensemble, an additional classifier, called the meta-classifier, has been introduced which uses the predictions of other sub-models to carry out further learning [49]. In general, ensemble models showed high classification performance and a significant improvement compared to base learners, results that are in line with the previous literature comparing boosting and logistic-regression-based classifiers [26].

## 4. Limitations and Strengths

There are some limitations of this study. First, although the patient database was divided into a “training set” and “testing set” through the K-fold cross-validation system (K = 10), which allowed us to carry out an evaluation independent from the generalization of the classifier model obtained during the training phase, the study was mono-institutional. An external validation is required to generalize our findings for a potential clinical application.

The results obtained were extracted following an exclusive quantitative evaluation of the texture characteristics of the images; integration with clinical data could help to implement classifier performances by increasing the accuracy and precision of the algorithms.

Furthermore, the selection of the ROIs was carried out manually, and, as reiterated several times previously, the sensitivity to the degree of signal intensity and the extension of the signal alterations are extremely variable intra- and inter-reader; the use of an automation system for the recognition of signal hyperintensities could overcome this problem.

Another limitation of our study was the use of panoramic STIR sequences of the WBMRI protocol; the use of dedicated coils for individual body areas could provide greater definition of the extension of the signal hyperintensities. In this study, however, we used WBMRI sequences because they are those currently used for the diagnosis and follow-up of patients with CNO. We therefore chose to extrapolate the bone marrow data from similar WBMRI examinations carried out on patients with suspected extra-skeletal diseases to make the data as comparable as possible. It would be appropriate to investigate whether the use of different sensitive fluid sequences can also provide and support similar Machine Learning results.

Therefore, we encourage future studies to evaluate CNO and all the pathologies that show “edema-like” alterations on T2-weighted/STIR sequences, providing a useful tool to support the clinician and radiologist to avoid overdiagnosis, delays in diagnosis and incorrect treatments for misinterpreted pathologies.

The strength of our study is precisely that we attempted to provide an automated tool for recognizing pathology and physiology to be introduced into clinical practice with the aim of speeding up the diagnostic process and avoiding the use of invasive techniques, often poorly accepted by pediatric patients, such as bone marrow biopsy.

## 5. Conclusions

Our results show the potential of Machine Learning and Deep Learning (DL) methods in discerning edema-like changes, particularly between CNO lesions and hematopoietic bone marrow changes in the pediatric population. The Neural Network classifier showed the overall best performances, while a Stacking classifier, based on Gradient Boosting and Random Forest as principal estimators and Logistic Regressor as final estimator, achieved the best results between classical ML methods.

## Figures and Tables

**Figure 1 diagnostics-14-00061-f001:**
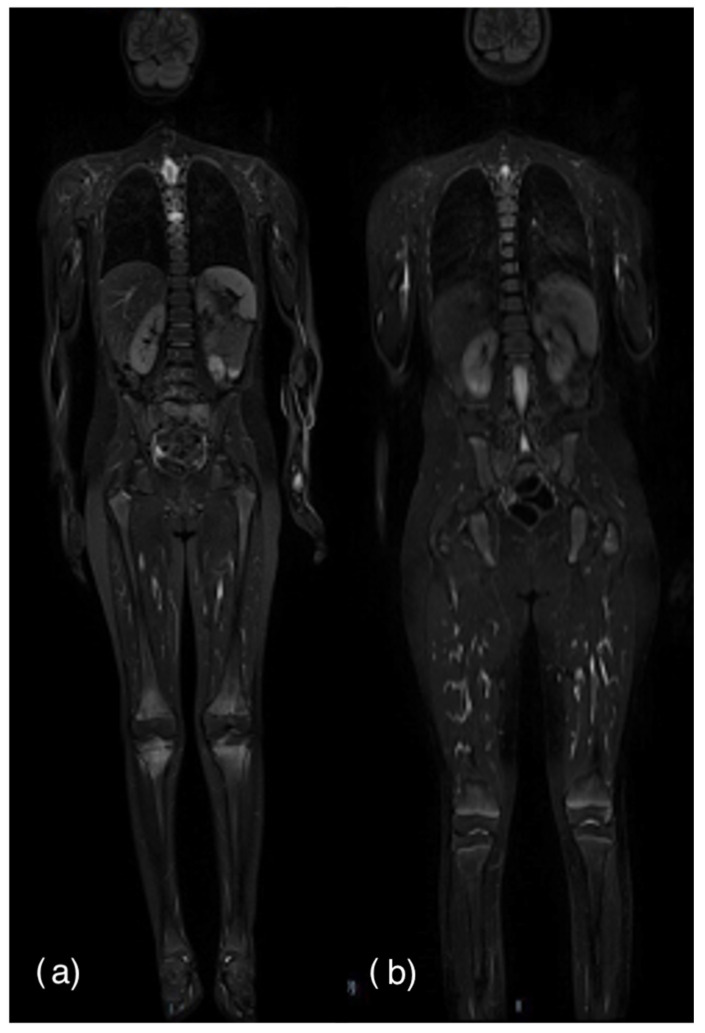
(**a**) WBMRI showing CNO hyperintense signal of femurs, tibias and a dorsal vertebrae. (**b**) WBMRI showing red marrow hyperintense signal of femurs, ilium and ischium.

**Figure 2 diagnostics-14-00061-f002:**
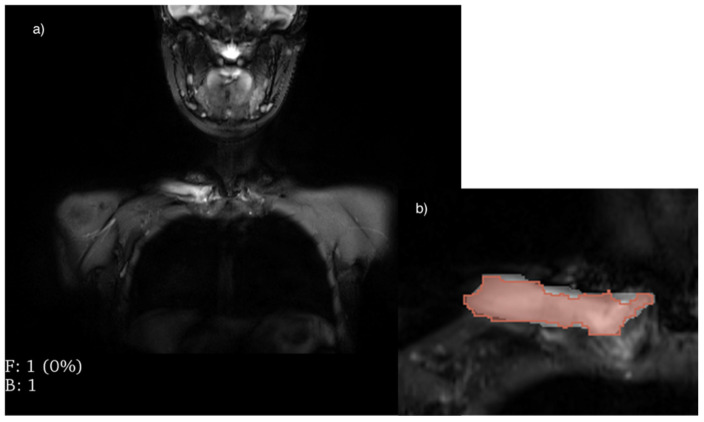
CNO hyperintensity sampling: (**a**) shows a hyperintense signal in the right clavicle, (**b**) shows the corresponding ROI drawn manually using the 3D Slicer program on the corresponding hyperintensity.

**Figure 3 diagnostics-14-00061-f003:**
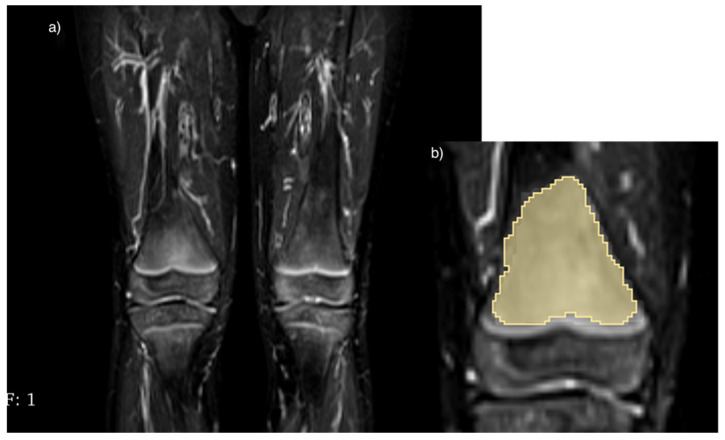
Red marrow hyperintensity sampling: (**a**) shows a hyperintense signal in the right femur, (**b**) shows the corresponding ROI drawn manually using the 3D Slicer program on the corresponding hyperintensity.

**Figure 4 diagnostics-14-00061-f004:**
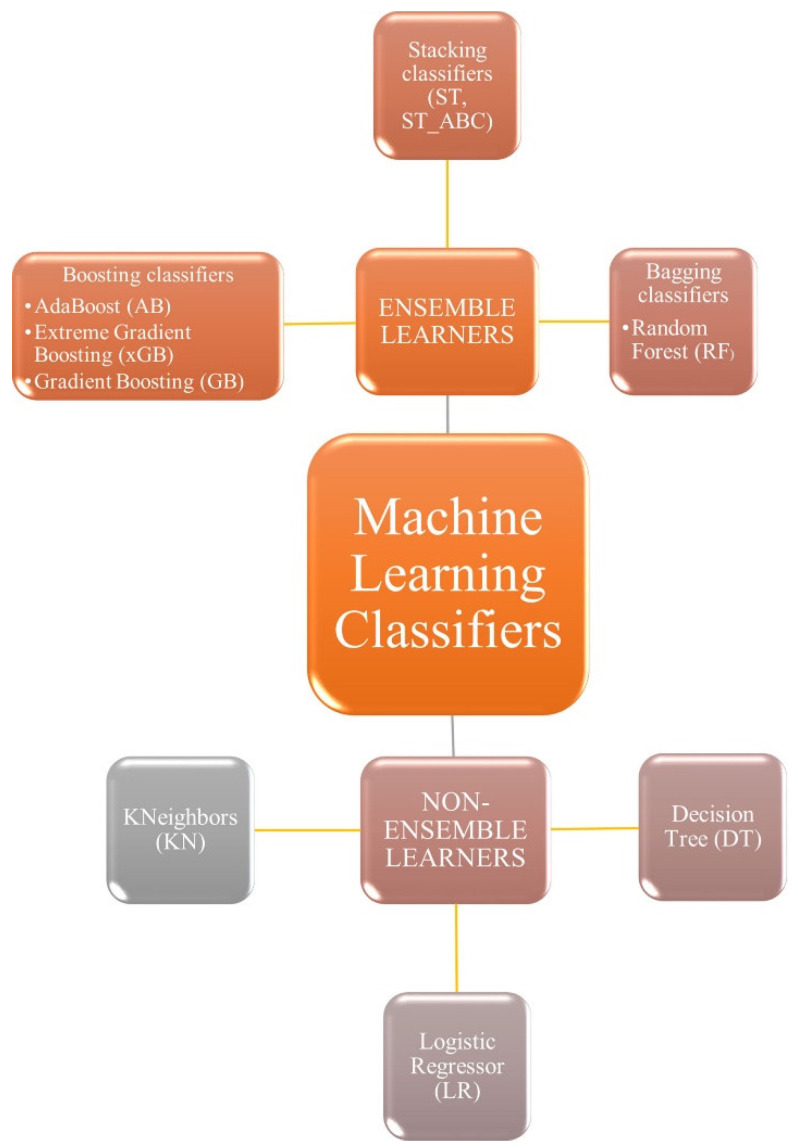
Machine Learning classifiers tested in the present study. Non-ensemble learners included KNeighbors, Logistic Regressor and Decision Tree. Ensemble learners included boosting, Stacking and bagging classifiers.

**Figure 5 diagnostics-14-00061-f005:**
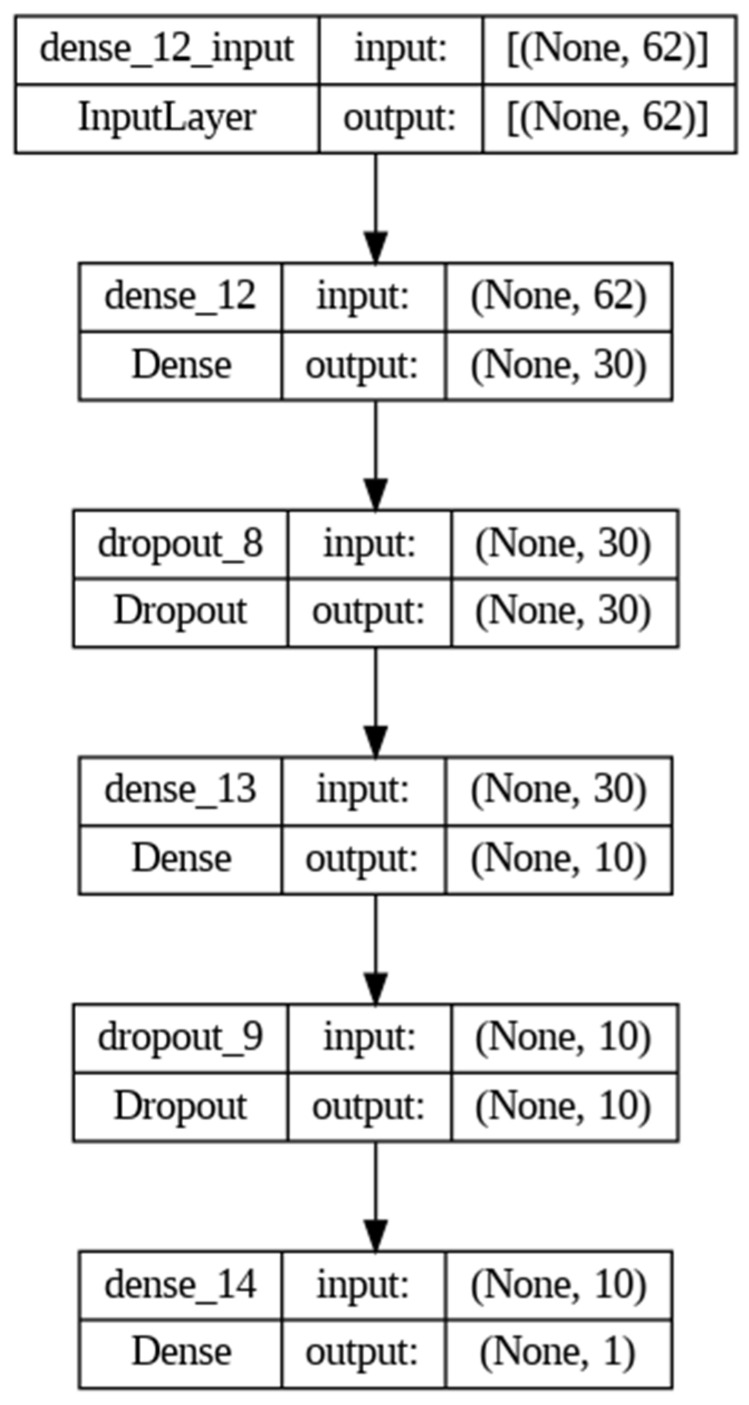
Neural Network structure: input layer with 62 units, first hidden layer with 30 units and L2 regularization, first dropout layer, second hidden layer with 10 units and L2 regularization, second dropout layer, output layer with 1 unit and sigmoid activation.

**Figure 6 diagnostics-14-00061-f006:**
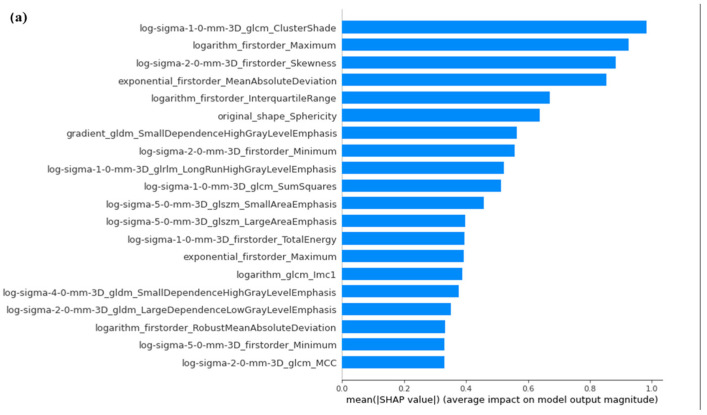
SHAP reporting of the 20 ranking significative features according to global (**a**) and local (**b**) point of view of the method adopted.

**Figure 7 diagnostics-14-00061-f007:**
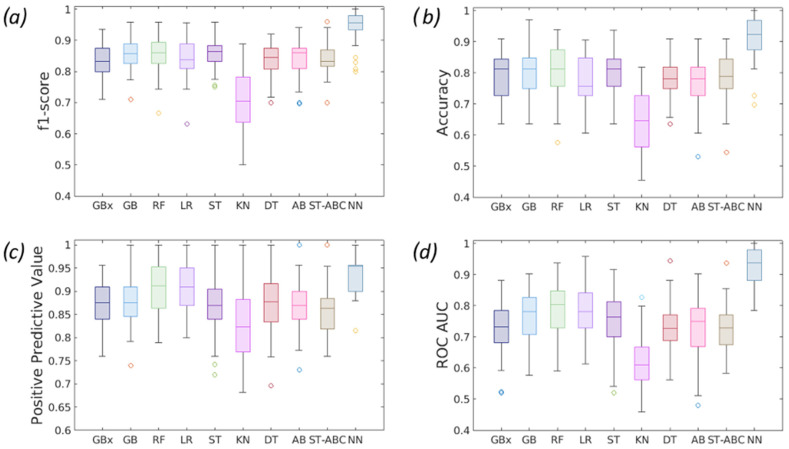
Boxplot comparison between the nine ML classifiers used in terms of (**a**) F1-score, (**b**) Accuracy, (**c**) Positive Predictive Value, (**d**) ROC/AUC.

**Table 1 diagnostics-14-00061-t001:** Accuracy, ROC-AUC, F1-score, Positive Predictive Value, MSE and RMSE values obtained for the ten classifiers.

	ACC	AUC	F1-Score	PPV	MSE	RMSE
GBx	0.84	0.83 with 95% CI 0.81–0.85	0.9	92%	0.2	0.4
GB	0.85	0.82 with 95% CI 0.8–0.84	0.9	91%	0.1	0.4
RF	0.84	0.82 with 95% CI 0.8–0.83	0.88	91.4%	0.2	0.4
LR	0.8	0.78 with 95% CI 0.76–0.81	0.85	90%	0.2	0.5
ST	0.85	0.81 with 95% CI 0.8–0.84	0.9	90%	0.2	0.4
KN	0.73	0.72 with 95% CI 0.7–0.74	0.8	87%	0.2	0.5
DT	0.78	0.74 with 95% CI 0.72–0.77	0.85	88.8%	0.2	0.5
AB	0.84	0.8 with 95% CI 0.77–0.82	0.88	89.8%	0.2	0.4
ST_AB	0.83	0.77 with 95% CI 0.74–0.79	0.89	88%	0.2	0.4
NN	0.91	0.93 with 95% CI 0.91–0.94	0.94	93.8%	0.2	0.4

## Data Availability

The datasets used and analyzed during the current study are available from the corresponding author on reasonable request.

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
