# Peer review of "Machine Learning Algorithm: Texture Analysis in CNO and Application in Distinguishing CNO and Bone Marrow Growth-Related Changes on Whole-Body MRI"

_diagnostics, 2023, doi:10.3390/diagnostics14010061_

Round 1
Reviewer 1 Report
Comments and Suggestions for Authors
Comments
In this manuscript, the authors analyzed the texture characteristics of the CNO bone lesions, identified as areas of altered intensity signal on STIR sequences, and distinguished them from the hyperintense signal of red bone marrow through Machine Learning Analysis.As authors proposed, the results show the potential of Machine Learning methods in discerning edema-like lesions. Here are some specific comments and suggestions:
1. For some figs like Figs.1 & 2, it is better to rearrange the layout of the article according to them. It will be more acceptable to compose two figs into one.And for Figs.3 & 4, you could provide enlarged images of the ROI areas.
2. You can provide as many comparison maps as possible regarding neural network recognition of disease areas.Moreover, you could show more works about comparing DL methods and others.There could be more charts to evidence your improvements.
3. For DL parts, it is suggested to provide the specific network structure used and provide pictures. If possible, further analyze whether the network structure is innovative in enhancing this work.
4. The resolution of some charts needs to be improved such like Fig.5.
5. Perhaps more about ‘A stacking classifier, based on Gradient Boosting and Random Forest as principal estimators and Logistic Regressor as final estimator, achieved the best results.’ can be explained through principles or charts
6. I would suggest the authors to provide a more detailed description of the extension and limitations of their proposed method in this work and if possible, in more aspects of medical image segmentation application. Specifically, it would be helpful to discuss how the proposed method could be further extended to other related applications or scenarios, as well as to acknowledge any potential limitations or drawbacks of the proposed approach. By addressing these points, the authors could provide a more comprehensive evaluation of their method and help readers better understand its potential strengths and weaknesses.
Comments on the Quality of English LanguageMinor editing of English language required.
Author Response
Here below you can find, in progressive numbers, the “original and reproduced” (in bold) comments raised by the referee, subdivided in numbers, followed by our replies to each point.
First of all, we would like to thank the reviewer for their valuable suggestions to improve our article and for the appropriateness of the comments raised.
In this manuscript, the authors analyzed the texture characteristics of the CNO bone lesions, identified as areas of altered intensity signal on STIR sequences, and distinguished them from the hyperintense signal of red bone marrow through Machine Learning Analysis.As authors proposed, the results show the potential of Machine Learning methods in discerning edema-like lesions. Here are some specific comments and suggestions:
- For some figs like Figs.1 & 2, it is better to rearrange the layout of the article according to them. It will be more acceptable to compose two figs into one.And for Figs.3 & 4, you could provide enlarged images of the ROI areas.
Reply: Thanks for pointing this out, we have edited the images as suggested.
- You can provide as many comparison maps as possible regarding neural network recognition of disease areas.Moreover, you could show more works about comparing DL methods and others.There could be more charts to evidence your improvements.
Reply: Thank you for your suggestion. We have included a wider description of the used models as well as pictures and graph. AB and GB are boosting methods, where AB reweights instances and GB excels with categorical data. xGB is highlighted as the fastest implementation of gradient boosted trees. RF is presented as a collection of decision trees for object classification based on attributes. DT serves as a basic algorithm for classification. LR is a widely used linear classifier for capturing linear relationships in data. Stacked Generalization in ST and ST_ABC is employed as an ensemble algorithm combining predictions from GB, RF, and LR. k-Nearest Neighbors (kNN) relies on data space distance for classification. NN are generally known to have very high performances related to image classification.
- For DL parts, it is suggested to provide the specific network structure used and provide pictures. If possible, further analyze whether the network structure is innovative in enhancing this work.
Reply: Thanks for pointing this out. We have included a wider description of the used models as well as pictures and graphs to show them. AB and GB are boosting methods, where AB reweights instances and GB excels with categorical data. xGB is highlighted as the fastest implementation of gradient boosted trees. RF is presented as a collection of decision trees for object classification based on attributes. DT serves as a basic algorithm for classification. LR is a widely used linear classifier for capturing linear relationships in data. Stacked Generalization in ST and ST_ABC is employed as an ensemble algorithm combining predictions from GB, RF, and LR. k-Nearest Neighbors (kNN) relies on data space distance for classification. NN are generally known to have very high performances related to image classification.
- The resolution of some charts needs to be improved such like Fig.5.
Reply: Thanks for pointing this out, we have edited the charts as suggested.
- Perhaps more about ‘A stacking classifier, based on Gradient Boosting and Random Forest as principal estimators and Logistic Regressor as final estimator, achieved the best results.’ can be explained through principles or charts
Reply: Thank you for your suggestion. The NN shows the best performances: ACC = 0.91, AUC = 0.93 with 95% CI 0.91-0.94, F1-score = 0.94, PPV = 93.8%. While among the classical ML classifiers Stacking classifier (ST) achieved the best results (ACC = 0.85, AUC = 0.81 with 95% CI 0.8-0.84, F1-score = 0.9, PPV = 90%). ST is an ensemble learner in which, compared to the bagging ensemble, an additional classifier, called meta-classifier, has been introduced, which uses the predictions of other sub-models to carry out further learning.49 In general ensemble models showed high classification performance and a significant improvement compared to base learners, results that are in line with previous literature comparing boosting and logistic regression-based classifiers.
- I would suggest the authors to provide a more detailed description of the extension and limitations of their proposed method in this work and if possible, in more aspects of medical image segmentation application. Specifically, it would be helpful to discuss how the proposed method could be further extended to other related applications or scenarios, as well as to acknowledge any potential limitations or drawbacks of the proposed approach. By addressing these points, the authors could provide a more comprehensive evaluation of their method and help readers better understand its potential strengths and weaknesses.
Reply: Thanks for pointing this out, very helpful comment. We tried to follow your suggestion by dedicating a specific paragraph in the discussion section where we highlight the strengths, limitations, and possible future directions of our work.
Reviewer 2 Report
Comments and Suggestions for Authors
This paper proposes an idea for applying machine learning methods to in discerning edema-like lesions, in particular by distinguishing CNO(Chronic Non-Bacterial Osteomyelitis) lesions from hematopoietic bone marrow changes in the pediatric population, and experiments are conducted to verify the potential of the method.
Strengths:
1. This paper proposed a potential method to distinguish CNO(Chronic Non-Bacterial Osteomyelitis) and bone marrow, which has the function of assisting medical diagnosis.
2. It provides an appropriate explanation for the approach and supports them through experiments.
Weaknesses:
1. This article mainly focuses on distinguish them from the hyperintense signal of red bone marrow through Machine Learning Analysis, but the title mentions artificial intelligence, which is too broad.
2. The author did not explain the motivation for their work and did not explain their contributions.
3. On page 2, line 26, “Recent studies have shown, …, to practicing radiologists”. The examples given are too thin to provide detailed explanations. And the examples are quite extensive, reflecting that artistic intelligence algorithms are often used in the field of radiological imaging, without focusing on the applications in CNO or bone marrow that you are analyzing. It is suggested that the authors provide more literature support and further explanation.
4. On page 7, line 13 of the “FEATURES SELECTION AND CLASSIFICATIONS”, the authors mention the use of the Boruta algorithm, a recently introduced feature selector method. But Boruta algorithm is a 2010 method, where is “recently introduced” reflected?
5. On page 7, “PATIENTS”, there is a bit of confusion in the selection datasets, compared to CNO patients, the second one is "for suspected subset of systemic extra skeletal disease". Why choose suspected patients instead of healthy ones? And do CNO patients themselves have red bone marrow features? So how can the red bone marrow characteristics of patients be distinguished from those caused by CNO? It is suggested that the authors provide more detailed explanations on the selection of datasets and annotate the sources of the data.
6. On page 7, the figure 5 is too small. It is recommended to place the two images separately and vertically. And the author did not explain the content of this picture,the meaning of the first 20 significative features.
7. Another question is, there is no comparative experiment in the experimental part. Is there any other method currently used to distinguish between CNO and bone marrow, or to recognize the features of these two?
8. In the discussion section, the logical structure is chaotic, the first half is more like the content of an introduction, introducing the research results of others in the field. It is suggested that the authors to reorganize the content of the discuss and introduction.
9. In the discussion section, line 61, “this minimal difference in intensity is perceptible to an expert eye” conflicts with " Due to the non-specificity of the symptoms, …, diagnosis is often difficult and delayed by about two years." in the line 6 of the introduction, and if the doctor is already easily distinguishable, the motivation for this article is weak. It is suggested that the author to Provide a clearer explanation of the motivation.
10. In the fourth paragraph of the results section, the evaluation metrics for each machine learning method can be presented in tabular form instead of a large textual narrative.
11. Authors should analyze the reasons why individual machine learning methods perform well or poorly and the factors contributing to the success of the proposed method.
12. The machine learning methods used in the paper are very early classical algorithms, and the authors may consider using recently proposed machine learning methods.
13. The authors should explain why 5-fold cross-validation was chosen instead of the more common 10-fold cross-validation.
14. The organization of the article needs further improvement, especially the introduction and discussion sections to Help readers understand the motivation of the article and existing research work in related fields.
15. There are several typos in the manuscript, please carefully proofread it.
16. Most of the cited papers in this paper are too early and the author should cite recent literature to prove the point.
Comments on the Quality of English LanguageExtensive editing of English language required.
Author Response
Here below you can find, in progressive numbers, the “original and reproduced” (in bold) comments raised by the referee, subdivided in numbers, followed by our replies to each point.
First of all, we would like to thank the reviewer for their valuable suggestions to improve our article and for the appropriateness of the comments raised.
This paper proposes an idea for applying machine learning methods to in discerning edema-like lesions, in particular by distinguishing CNO(Chronic Non-Bacterial Osteomyeliti) lesions from hematopoietic bone marrow changes in the pediatric population, and experiments are conducted to verify the potential of the method.
Strengths:
- This paper proposed a potential method to distinguish CNO(Chronic Non-Bacterial Osteomyelitis)and bone marrow, which has the function of assisting medical diagnosis.
- It provides an appropriate explanation for the approach and supports them through experiments.
Weaknesses:
- This article mainly focuses on distinguish them from the hyperintense signal of red bone marrow through Machine Learning Analysis, but the title mentions artificial intelligence, which is too broad.
Reply: Thanks for pointing this out. We have changed the title to be as specific as possible with “MACHINE LEARNING ALGORITHM: TEXTURE ANALYSIS IN CNO AND APPLICATION IN DISTINGUISHING CNO AND BONE MARROW GROWTH-RELATED CHANGES ON WHOLE BODY MRI”
- The author did not explain the motivation for their work and did not explain their contributions.
Reply: Thanks for pointing this out, we have modified the introduction to try to better motivate our study, in particular by adding the phrase “We have attempted to provide clinical help for the diagnosis of CNO in order to avoid overdiagnosis and consequent overtreatment of the pathology, and to avoid invasive procedures such as bone biopsy.”
- On page 2, line 26, “Recent studies have shown, …, to practicing radiologists”. The examples given are too thin to provide detailed explanations. And the examples are quite extensive, reflecting that artistic intelligence algorithms are often used in the field of radiological imaging, without focusing on the applications in CNO or bone marrow that you are analyzing. It is suggested that the authors provide more literature support and further explanation.
Reply: Thanks for pointing this out, the literature cited is generic because now there are no studies that focuses on the differentiation between the two types of hyperintensity (CNO e bone marrow) through machine learning or that use AI to study bone lesions, reference "25" is the one that comes closest to the topic discussed. However, we have added in the text a reference regarding an automatic segmentation method of hyperintense signal of CNO and bone marrow.
- On page 7, line 13 of the “FEATURES SELECTION AND CLASSIFICATIONS”, the authors mention the use of the Boruta algorithm, a recently introduced feature selector method. But Boruta algorithm is a 2010 method, where is “recently introduced” reflected?
Reply: Thanks for pointing this out. We corrected the text by deleting “recently introduced”.
- On page 7, “PATIENTS”, there is a bit of confusion in the selection datasets, compared to CNO patients, the second one is "for suspected subset of systemic extra skeletal disease". Why choose suspected patients instead of healthy ones? And do CNO patients themselves have red bone marrow features? So how can the red bone marrow characteristics of patients be distinguished from those caused by CNO? It is suggested that the authors provide more detailed explanations on the selection of datasets and annotate the sources of the data.
Reply: Thanks for pointing this out, for bone marrow data sampling we chose patients who underwent whole body MRI due to suspicion of extra skeletal disease, for example dermatomyositis, in order to be sure of not having bone involvement that could distort the data. Being a retrospective study, we did not enrol healthy volunteers, also because it is a study conducted on paediatric patients, often under general anaesthesia. The WBMRIs of the CNO group were all performed on patients who had histological confirmation of the disease as expressed in the article. Your question "So how can the red bone marrow characteristics of patients be distinguished from those caused by CNO?" This is precisely the key point of our study. Our group of patients with CNO had histological diagnoses of CNO. The aim of our study is precisely to be able to answer your question through an automatic recognition method after training on certain data. We have reorganized the "PATIENTS" section trying to make the choice of data selection more understandable.
- On page 7, the figure 5 is too small. It is recommended to place the two images separately and vertically. And the author did not explain the content of this picture,the meaning of the first 20 significative features.
Reply: Thank you very much, we have displaced the images as suggested. The result of the SHAP analysis reports the 20 ranking significative features according to global and local point of view of the method adopted. The features highlighted in the image are the ones which contribute the most to the prediction and therefore might be more specific in the characterization of the disease.
- Another question is, there is no comparative experiment in the experimental part. Is there any other method currently used to distinguish between CNO and bone marrow, or to recognize the features of these two?
Reply: Thanks for pointing this out, currently, there are some clinical scores for diagnosis of CNO that we have add in the introduction section whit the references, but these scores are no validated. There are no automatic recognition methods for distinguishing the two types of signals hyperintensity. The only method used is based on the "eye" of the radiologist, a subjective method that can provide many false positives. This is one of the reasons why we tried to standardize the diagnosis through the use of an automatic lesion recognition via machine learning.
- In the discussion section, the logical structure is chaotic, the first half is more like the content of an introduction, introducing the research results of others in the field. It is suggested that the authors to reorganize the content of the discuss and introduction.
Reply: Thanks for pointing this out, very helpful comment. We have reorganized the content of the discuss and introduction.
- In the discussion section, line 61, “this minimal difference in intensity is perceptible to an expert eye” conflicts with " Due to the non-specificity of the symptoms, …, diagnosis is often difficult and delayed by about two years." in the line 6 of the introduction, and if the doctor is already easily distinguishable, the motivation for this article is weak. It is suggested that the author to Provide a clearer explanation of the motivation.
Reply: Thanks for pointing this out, we can define "expert eye" of a radiologist who deals with paediatric and/or rheumatological pathology. Radiologists without these hyper specialties may find difficulty to distinguish between pathology and physiology. We have modified the manuscript by providing examples of the variability of the degree of signal hyperintensity of the bone marrow and of the difficulty faced daily in distinguishing between physiological and pathological.
- In the fourth paragraph of the results section, the evaluation metrics for each machine learning method can be presented in tabular form instead of a large textual narrative.
Reply: Thank you very much for the advice. We have implemented your suggestion and the results are much more visually immediate as well as more readable.
- Authors should analyze the reasons why individual machine learning methods perform well or poorly and the factors contributing to the success of the proposed method.
Reply: Thank you for your suggestion. We have provided a more detailed insight of the proposed methods. AB and GB are boosting methods, where AB reweights instances and GB excels with categorical data. xGB is highlighted as the fastest implementation of gradient boosted trees. RF is presented as a collection of decision trees for object classification based on attributes. DT serves as a basic algorithm for classification. LR is a widely used linear classifier for capturing linear relationships in data. Stacked Generalization in ST and ST_ABC is employed as an ensemble algorithm combining predictions from GB, RF, and LR. k-Nearest Neighbors (kNN) relies on data space distance for classification. NN are generally known to have very high performances related to image classification. The results mirror the description of each classificator with NN showing the best performances: ACC = 0.91, AUC = 0.93 with 95% CI 0.91-0.94, F1-score = 0.94, PPV = 93.8%. Among the classical ML classifiers Stacking classifier (ST) achieved the best results (ACC = 0.85, AUC = 0.81 with 95% CI 0.8-0.84, F1-score = 0.9, PPV = 90%). ST is an ensemble learner in which, compared to the bagging ensemble, an additional classifier, called meta-classifier, has been introduced, which uses the predictions of other sub-models to carry out further learning.49 In general ensemble models showed high classification performance and a significant improvement compared to base learners, results that are in line with previous literature comparing boosting and logistic regression-based classifiers.
- The machine learning methods used in the paper are very early classical algorithms, and the authors may consider using recently proposed machine learning methods
Reply: Thank you for your suggestion. We have added a simple Neural Network (NN) model to the previously used classifiers. The NN, implemented using the Keras library, is especially designed for binary classification tasks with 2 hidden layers and an output layer with a sigmoid activation; L2 regularization and dropout are applied in the hidden layers to prevent overfitting. The results obtained by the NN outdid the other classical ML classifiers with ACC = 0.91, AUC = 0.93 with 95% CI 0.91-0.94, F1-score = 0.94, PPV = 93.8%, confirming NNs strong performances in image classification, which showcase robust performance and adaptability across diverse datasets.
- The authors should explain why 5-fold cross-validation was chosen instead of the more common 10-fold cross-validation.
Reply: Thank you very much for pointing this out. 5-fold cross-validation is actually a typing mistake as the actual k-fold used is 10 with five random states (23, 35, 42, 56, 67).
- The organization of the article needs further improvement, especially the introduction and discussion sections to Help readers understand the motivation of the article and existing research work in related fields.
Reply: Thanks for pointing this out, very helpful comment. We have reorganized both the introduction and the discussion to try to emphasize our intent. Furthermore, in the discussion section we have created a specific paragraph in which we highlight the strengths, limitations and possible future directions of our work.
- There are several typos in the manuscript, please carefully proofread it.
Reply: Thanks for pointing this out, we corrected it.
- Most of the cited papers in this paper are too early and the author should cite recent literature to prove the point.
Reply: Thanks for pointing this out, we have carried out a new bibliographical review and we have included very recent articles.
Round 2
Reviewer 1 Report
Comments and Suggestions for Authors
After modification, the article has a lot of ascension.
Reviewer 2 Report
Comments and Suggestions for Authors
Accept in present form